# Dietary Relationship with 24 h Urinary Iodine Concentrations of Young Adults in the Mountain West Region of the United States

**DOI:** 10.3390/nu12010121

**Published:** 2020-01-01

**Authors:** Demetre E. Gostas, D. Enette Larson-Meyer, Hillary A. Yoder, Ainsley E. Huffman, Evan C. Johnson

**Affiliations:** 1Department of Family and Consumer Sciences, University of Wyoming, Laramie, WY 82071, USA; gostas1@uwyo.edu; 2Division of Kinesiology, University of Alabama, Tuscaloosa, AL 35487, USA; hayoder@crimson.ua.edu; 3University of Utah School of Medicine; Salt Lake City, UT 84108, USA; ainsley.huffman@hsc.utah.edu; 4Division of Kinesiology & Health, University of Wyoming; Laramie, WY 82070, USA; ejohns54@uwyo.edu

**Keywords:** Iodine Status, Food Frequency Questionnaire, iodized salt, iodine intake, dairy intake, adults

## Abstract

Background: Iodine deficiency is not seen as a public health concern in the US. However certain subpopulations may be vulnerable due to inadequate dietary sources. The purpose of the present study was to determine the dietary habits that influence iodine status in young adult men and women, and to evaluate the relationship between iodine status and thyroid function. Methods: 111 participants (31.6 ± 0.8 years, 173.2 ± 1.0 cm, 74.9 ± 1.7 kg) provided 24 h urine samples and completed an iodine-specific Food Frequency Questionnaire (FFQ) for assessment of urinary iodine content (UIC) as a marker of iodine status and habitual iodine intake, respectively. Serum Thyroid Stimulating Hormone (TSH) concentration was evaluated as a marker of thyroid function. Spearman correlational and regression analysis were performed to analyze the associations between iodine intake and iodine status, and iodine status and thyroid function. Results: 50.4% of participants had a 24 h UIC < 100 µg/L). Dairy (r = 0.391, *p* < 0.000) and egg intake (r = 0.192, *p* = 0.044) were the best predictors of UIC, accounting for 19.7% of the variance (*p* ≤ 0.0001). There was a significant correlation between UIC and serum TSH (r = 0.194, *p* < 0.05) but TSH did not vary by iodine status category (F = 1.087, *p* = 0.372). Discussion: Total dairy and egg intake were the primary predictors of estimated iodine intake, as well as UIC. Iodized salt use was not a significant predictor, raising questions about the reliability of iodized salt recall. These data will be useful in directing public health and clinical assessment efforts in the US and other countries.

## 1. Introduction

Iodine is an essential trace mineral that forms the building blocks of the thyroid hormones thyroxine and triiodothyronine, which are critical regulators of metabolic activity [1]. Iodine’s major environmental source is the ocean [2], with seafood and seaweed providing significant dietary sources. The iodine content of most other foods, however, is low and dependent on soil content and agriculture practices [3]. Exceptions include dairy products, which may be richer sources due to livestock iodine supplementation and use of iodophors for cleaning milk udders [4,5,6]. Insufficient iodine intake leads to iodine deficiency, which can manifest as hypothyroidism and endemic goiter in adults [3,7]. Iodine deficiency is of particular concern in women of reproductive age, as many pregnancies are unplanned [8], and deficiency can impair fetal cognitive and physical development [3,7].

Although iodine deficiency is a worldwide concern, with nearly one-third of the global population thought to be deficient [9], the iodine status of the US population has been viewed as adequate since the widespread iodization of salt in the late 1920s [10]. Urinary iodine concentration (UIC) measured over 24 h is a commonly used biomarker to assess iodine status in populations [3,11]. The most recent data from the National Health and Nutrition Examination Survey (NHANES 2005–2006 and 2007–2008) [12] show median UIC as adequate at 164 µg/L, with just under 10% of the population categorized as severely deficient [12]. Still, some subpopulations of the US may be vulnerable to deficiency due to food selection patterns or avoidance of iodized salt. These at-risk subpopulations include vegans/vegetarians [13], those who avoid seafood and/or dairy [14], and those follow a sodium restricted diet [15,16,17] or eat local foods in regions with iodine-depleted soils [2,3]. The Institute of Medicine [18] and the American Heart Association [19] have advocated for decreasing sodium intake to less than 2300 mg per day [18] and, more prudently, to less than 1500 mg per day [19], which could be reducing Americans’ intake of iodized salt. Additionally, apparent trends toward local and plant-based diets may negatively influence iodine status depending on food selection patterns and habits (e.g., avoidance of seafood and dairy) and the content of local soils. Therefore, despite the labeling of the US populations’ iodine intake as adequate, certain dietary choices, including strict adherence to dietary recommendations to restrict salt intake [20], may directly influence iodine status and indirectly influence thyroid function.

The present study was a pilot study that aimed to assess iodine intake and status in a sample of young adult men and women and determine the dietary patterns and habits that influence the observed iodine status. Twenty-four hour UIC was used as a reference standard for estimation of iodine status [7,21] although there is no consensus on the biomarker to use for the assessment of individual iodine status [22,23]. We hypothesized, based on the above, that we would observe at least some individuals with a low 24 h UIC (<100 µg/L), and that UIC would be associated with dietary factors such as the frequency of milk, fish and seafood intake, multivitamin and iodized salt use, and vegetarian status. A secondary purpose was to explore the relationship between iodine status and hypothyroidism using the thyroid stimulating hormone (TSH) as a general marker of thyroid function.

## 2. Materials and Methods 

This study was conducted concurrently as part of a larger study evaluating urine color as a marker of change in daily water intake [24] conducted over a 13 month period beginning in the spring of 2016. Male and female participants between the age of 18 and 45 years were recruited from the Laramie, Wyoming community. To be eligible, participants had to be in good overall health and not have a health condition that could influence study results (e.g., anemia, diabetes, cystic fibrosis, cancer autoimmune disorders). Exclusionary criteria are previously published [24] and include: inability to understand and write English (for ability to complete written survey instruments); evidence of clinically relevant metabolic, cardiovascular, hematologic, hepatic, gastrointestinal, renal, pulmonary, endocrine or psychiatric history of disease (based on the medical history questionnaire); pregnancy or breast-feeding; regular prescription drug treatment within 15 days prior to start of the study; inability to discontinue use of specific dietary/herbal supplements (calcium, chromium, vitamin C, cat’s claw, chaparral, cranberry, creatine, ephedra, germanium, hydrazine, licorice, l-lysine, pennyroyal, thunder god vine, willow bark, wormwood oil, yellow oleander, yohimbe); currently exercising >4 h per week; changes in diet or in body mass of >2.5 kg (~5 lbs) in the past month; or recent relocation from low altitude to Laramie within the past three months. The study was approved by the Institutional Review Board of the University of Wyoming. Volunteers were informed that the urine samples collected to assess hydration status would also be used to establish the iodine status of a healthy population of individuals through measurement of iodine levels in urine and if these levels are related to specific dietary or lifestyle factors. They were also informed of any possible risks prior to giving written formal consent to participate in the study. 

### 2.1. Overview of Testing

This analysis was performed on 111 of the 125 total participants enrolled in the study, who had complete data on 24 h urinary iodine concentrations (UIC) and valid responses from a food frequency questionnaire (FFQ). Reasons for exclusion of 14 of the 125 participants included: voluntary withdrawal prior to completion, not following study protocol instructions, not providing adequate urine for analysis or not turning in complete dietary data. At study initiation (baseline), participants completed a dietary habits survey and an iodine-specific food-frequency questionnaire (FFQ). Height and weight were measured on a stadiometer (Health o meter ^®^, Model 201 HR) and digital weight scale (Seca, Model 780 2321138), respectively, with body mass index (BMI) calculated as kg/m^2^. Physical activity was estimated using the International physical Activity Questionnaire (IPAQ) [25]. 

### 2.2. Measurement of Urinary Iodine Concentration and Iodine Status 

As part of an 11 day collection protocol, participants collected a 24 h urine sample for the purpose of UIC measurement on the morning of day two. Participants were asked to void and discard the first morning urine sample and then collect all subsequent samples for 24 h, ending with the first sample upon waking on day three. Food and fluid intake were ad libitum during this 24 h urine collection. Iodine in urine was measured by a commercial laboratory (Mayo Clinic, Rochester, MN) using an inductively coupled plasma-mass spectrometry using tellurium as an internal standard and an aqueous acid calibration. The repeated tolerance acceptability was 10 ng/mL or 10%. Twenty-four hour UIC was defined using the following criteria: <20 µg/L (severe iodine deficiency), 20–49 µg/L (moderate iodine deficiency), 50–99 µg/L (mild iodine deficiency), 100–199 µg/L (adequate iodine nutrition), 200–299 µg/L (more than adequate iodine intake), ≥300 µg/L (excessive iodine intake) [15]. UIC was also entered into the equation of Zitterman, which incorporates body mass to estimate intake [26,27]. Urinary iodine (µg/L) × 0.0235 × body weight (kg) = Daily Iodine Intake.

### 2.3. Iodine Intake, Frequency of Iodine Containing Foods and Dietary Habits 

The iodine-specific FFQ (Appendix B) evaluated the frequency of consumption of 43 food items known to have significant iodine content (e.g., seafood, seaweed, dairy—see Table 1). Frequency was evaluated according to the following responses: (a) never or less than one time per month; (b) one to three times per month; (c) one time per week; (d) two to four times per week; (e) five to six times per week; (f) one time per day; (g) two to three times per day; (h) four to five times per day; (i) or six or more times per day). Daily intake of iodine was estimated by multiplying the frequency midpoint by the average content of each vitamin D-containing food and expressed as IU/day (assuming 30 days per month), as previously outlined by Halliday et al. [28]. As iodine content is not available in food composition tables or databases, iodine content of the food items in the FFQ was derived from several sources (Table 1), with a majority of the data coming from the ongoing Total Diet Study (TDS) [29]. Iodine content in the TDS is listed per 100 g of the selected food item. Iodine content was recalculated from the TDS to iodine per serving size, to match the household measured listed in the FFQ. The iodine content in other sources was also converted to adjust given units to iodine per serving size. For ease of analysis, iodine intake from specific categories of foods were combined, which included estimates of iodine intake from total dairy (fluid milk plus yogurt), total fish (all types of fish) and total seafood (total fish plus all types of shellfish). 

The Dietary Habits Survey (Appendix C) addressed questions regarding the frequency of table salt use in salting and cooking foods, and the type of table salt typically consumed (iodized, non-iodized). It also addressed whether participants followed a vegetarian diet, frequented a farmer’s market for local food purchases, were a member of a Community Supported Agriculture program (CSA), or maintained a home garden for growing food, based on Yes, Sometimes, or NO responses (Appendix C). Both instruments were developed for the current study and have not yet been validated. 

### 2.4. Assessment of Thyroid Function

A blood sample was obtained on the morning of day three for measurement of TSH via immunometric assay, (Regional West, Scotts Bluff, NE). Daily quality control at different representative levels was evaluated across the measurement range, as precision changed with concentration for 112 replicates. The CV was 3.4% when the mean was 50 µg I per 24 h, and 1.3% when the mean was 191 µg I per 24 h. The presence of hypothyroidism (TSH > 4.68 mIU/L) and hyperthyroidism (TSH < 0.47 mIU/L) was evaluated based on standard laboratory ranges (normal TSH = 0.47 to 4.68 mIU/L) as well as using recent, more conservative criteria for subclinical hypothyroidism (TSH > 2.5) [36]. 

### 2.5. Statistical Analysis

Data were analyzed using IBM SPSS statistics software (SPSS Inc., Chicago, IL; version 24.0). Analysis of Variance (ANOVA) was used to compare differences in iodine intake, iodine status, thyroid function and other key variables by sex (male vs. female). Correlation coefficients (Pearson or Spearman Rank) were used to evaluate the associations between iodine status (i.e., UIC) and gross iodine intake, or iodine intake from specific foods or supplements (dairy, seafood, seaweed, iodized salt, multivitamin intake, etc.). Pearson correlations were also used for the evaluation of UIC and serum TSH relationships. In most cases, Spearman Rank Coefficients were used instead of Pearson Correlation Coefficients, due to the general non-normal distribution of intake data. Multiple linear Regression Models (Backwards Regression) were created to determine which dietary source(s) were the largest contributors to iodine intake and iodine status. ANOVA was also used to determine whether there were differences in iodine intake and status by the items indicated on the dietary habits survey, which included following a vegetarian diet, frequenting farmer’s markets for local food purchases, being CSA members or maintaining a home garden for growing food. Data are expressed as means ± SEM unless otherwise specified. Significance was set as an alpha < 0.05. 

## 3. Results

### 3.1. Subject Characteristics

The characteristics of the 111 participants are shown in Table 2. BMI and reported physical activity varied widely among participants. Men were taller, heavier, and had a higher BMI than women (*p* < 0.05). The sample size is provided in parenthesis in the case of occasional missing datapoints, attributed to inadequate samples for analysis or missing responses on questionnaires.

### 3.2. 24 h UIC and Iodine Status

Twenty-four hour UIC ranged between 15 and 714 µg/L. The median value was 98.0 µg/L (interquartile range = 60.0–180.0), with no difference by sex (*p* = 0.36). The frequency of the various categories of iodine status based on WHO criteria is shown in Figure 1.

Body Mass Index (BMI) and International Physical Activity Questionnaire (IPAQ) [37] scales were used in classifications. BMI classifications are as follows: Underweight (<18.5), Normal weight (18.5–24.9), Overweight (25.0–29.9), Obese (>30.0). IPAQ classifications are as follows: Low (not meeting criteria for Moderate or High), Moderate (likely doing 30 min moderate intensity physical activity on most days), High (likely doing at least 1 h of moderate intensity physical activity per day) [25]. 

### 3.3. Estimated Daily Iodine Intake and Frequency of Intake of Iodine-Containing Foods 

Frequency of consumption of selected iodine-containing foods along with supplements is shown in Table 3. Overall, estimated daily iodine intake ranged from 36.4 to 1113.3 µg/day. Daily intake averaged 327.7 µg (SD: 21.0; median 291.4) and was not different by sex (*p* = 0.49). While the mean and median iodine intake was higher than the U.S. Recommended Dietary Allowance (RDA) for adults of 150 µg/day, 21.6% (*n* = 24) had estimated intakes less than the RDA, and 1.8% (*n* = 2) had intakes greater than the Upper Limit of 1100 µg/day. The estimated average daily iodine intakes from contributing foods are as follows) total dairy (100.0 µg ± 8.6), eggs (18.7 µg ± 2.3), total fish (4.4 µg ± 1.3), total seafood (includes fish plus shellfish; 8.4 µg ± 1.7), iodized table salt (33.7 µg ± 5.1), multivitamin (47.61 µg ± 9.0), and seaweed (0.81 µg ± 0.25). Using Spearman rank correlation coefficients, the total estimated iodine intake from the FFQ was correlated with reported total milk intake (r = 0.769, *p* < 0.001), dairy intake (milk plus yogurt) (r = 0.716, *p* < 0.01), egg consumption (r = 0.295, *p* = 0.002), total fish intake (r = 0.191, *p* = 0.044) and multivitamin use (r = 0.460, *p* < 0.01) but not with total seafood consumption (r =0.169, *p* = 0.08), seaweed consumption (r = −0.050, *p* = 0.6), or iodized salt intake (r = 0.141, *p* = 0.14).

A linear regression model was created to determine which food source was the biggest determinant of estimated iodine intake. Dietary sources that were significantly correlated with total intake by simple correlation analysis (total dairy, egg consumption, total fish and multivitamin use) were entered, along with iodized table salt and seaweed. As shown in Table 4, all sources remained significant contributors to estimated iodine intake.

### 3.4. Relationship between Iodine Intake and Iodine Status 

Using Spearman Rank Correlations, the estimated total iodine intake from the FFQ was correlated with UIC (r = 0.310, *p* = 0.001) (Figure 2a). FFQ-estimated iodine intake was also correlated with predicted iodine intake using the equation of Zimmerman UIC (r = 0.327, *p* = 0.001) which incorporates UIC and body mass (Figure 2b) [26,27].

By Spearman Rank Correlation Coefficients, UIC was correlated with reported total dairy intake (r = 0.391, *p* < 0.01), egg consumption (r = 0.192, *p* = 0.044) and seaweed consumption (r = −0.239, *p* = 0.011), but not with total fish intake (r = −0.003, *p* = 0.973), total seafood consumption (r = −0.055, *p* = 0.569), iodized table salt use (r = 0.044, *p* = 0.646) or multivitamin use (r = 0.031, *p* = 0.749). 

A linear regression model was created to determine which food source(s) were the biggest determinant of 24 h UIC. Intake of iodine from dairy products, eggs, seaweed, fish, iodized salt and eggs were entered into the model. As shown in Table 5, only dairy and eggs remained significant contributors to UIC, explaining 19.7% of the variance (*p* ≤ 0.0001). Total fish, iodized table salt, multivitamin use and seaweed consumption accounted for an additional 3.2% of the variance, but were not significant contributors (*p* > 0.10). 

### 3.5. Dietary Habits, Iodine Intakes and Intake Status 

Twenty-one participants neglected to complete all questions on the Dietary Habits Survey. Complete data for those responding to questions addressing vegetarian status (*n* = 94), frequently visiting a farmer’s market or food co-op for local food purchases (*n* = 95), having membership in a CSA (*n* = 94), and growing food in a home garden (*n* = 95) are shown in parentheses. Of those who completed the questions, five reported following a vegetarian or vegan diet, 49 reported regularly or sometimes frequenting a farmer’s market for local food purchases, with 37 being members of a CSA or maintaining a home garden for growing food. Differences between estimated iodine intake or UIC were not observed according to whether participants reported following a vegetarian diet (*p* = 0.52 and 0.91), regularly or sometimes frequenting a farmer’s market (*p* = 0.60 and 0.49), being a CAS member (*p* = 0.44 and 0.38) or maintaining a home garden (0.58 and 0.49) (*p* > 0.05). 

### 3.6. Relationship between Iodine Status and Thyroid Function

The mean TSH (*n* = 107) was 2.07 ± 0.11 mIU/L. Based on standard laboratory criteria, four participants were classified with hypothyroidism (TSH > 4.68 mIU/L) and one was classified with hyperthyroidism (TSH < 0.47 mIU/L). Using more conservative criteria (TSH > 2.5), 24 were considered to have an underactive thyroid [36]. By Spearman rank correlation coefficients there was a correlation (*p* < 0.05) between serum TSH concentration and UIC (r = 0.194, *p* = 0.045). TSH concentration, however, did not differ by WHO iodine deficiency categories (*p* = 0.372). Notably, no differences in iodine intake or status were observed in those with TSH indicative of hypo- or hyperthyroid. 

## 4. Discussion

The overall purpose of this pilot study was to evaluate iodine intake and status in a sample of young adult men and women and determine if intake and UIC, as a biomarker of status, are influenced by dietary patterns. Overall, we found that our sample, with a median UIC of 98.0 µg/L and with 16.2% with a UIC below <50 µg/L, may be borderline deficient. However, there was high intersubjective variability, providing an opportunity to explore dietary relationships with UICs. In contrast, 23.4% of participants had a 24 h UIC of >200 µg/L. These data suggest that some members of our population may be at risk of compromised or excessive iodine status if such patterns are consistently observed (e.g., are not based on a single 24 h UIC). Both inadequate and excess iodine intake can be linked to adverse health consequences [15,38]. These findings, however, are relevant only to the sample examined and cannot be extrapolated to, and are not necessarily representative of, the young adults in our town, the mountain west region or elsewhere in the US. However, the high intersubjective variability provided the opportunity to explore dietary relationships with UIC to estimate the dietary factors that may place individuals at risk of a single suboptimal 24 h UIC. 

Total dairy and egg intake were the primary predictors of estimated iodine intake as well as UIC. Importantly, these two foods together predicted approximately 20% of the variance in UIC (Table 5). These results are in agreement with previous research showing a link between dairy and egg consumption and iodine status [39,40]. Dairy has been determined to be a reliable source due to iodine-fortified cattle feed [14] (which influences circulating iodine and the uptake of iodine by mammary epithelial cells and secretion into milk) [41] and the use of iodophor disinfecting agents on udders, although this can vary across the seasons and with industry practices [5,14,42]. In agreement with our study, a published review has suggested that milk and dairy contribute ~13% to 64% of the recommended daily iodine intake based on country-specific food intake data [6]. Eggs draw their iodine content from the laying hen’s feed [43,44,45,46], which is increased with iodine supplementation. Dairy and eggs might be easier to recall compared to other sources such as iodized table salt, seafood or fish. Also, dairy is often a habitual food item, with yogurts and milk often complimenting popular breakfast and snacks such as ready-to-eat cereal with milk, yogurt with fruit, coffee with milk or a protein shake mixed with milk. Additionally, since eggs must be cracked or consumed individually, the physical nature of preparation could make serving frequency and size easier to recall. 

In contrast, iodine intake from iodized table salt consumption was not found to be a primary predictor of iodine status within the present study. This finding is contrary to our hypothesis, as iodized salt is among the densest iodine sources, with more than threefold the amount of iodine per serving compared to an egg. This may be due to several factors. People may not know what iodized salt is or be aware of which sources of salt are iodized. This includes salt found in the kitchen, used in restaurants or used in processed foods. Iodization of salt, which was initiated in the United States in the late 1920s [10], is not currently mandated by the Food and Drug Administration. It is estimated that only 20% of the salt consumed in the United States is iodized [35]. Error may have been introduced if participants assumed that salt in processed foods or all salt in the saltshakers at restaurants is iodized. Additionally, it may be objectively difficult for participants to recall iodized salt use over a three month period. Salt is a seasoning that may be used sporadically in cooking and when salting foods to taste and, under these conditions, may contribute little to overall dietary intake. Finally, the amount of iodine found in iodized salt may vary considerably and depend on how long the salt has been on the grocery store shelf and in individual cupboards. One study found that iodine content varied among different brands of salt, and even within a single container [35]. Several others found that the iodine content of foods cooked with iodized salt depends on the food type and cooking method [47,48]. The increased popularity of blended salts (i.e., garlic salt) and artesian salts (i.e., Pink Himalayan, sea salt and Kosher rock salt) add another element of uncertainty as these products are typically not iodized [35], yet often share the same shelf space as iodized salt. As it is likely that habitual use of iodized salt does contribute to iodine status [49], future versions of our FFQ will consider the recall time of our FFQ as well as use of a photographic album during FFQ administration, along with more detailed and explicit questions related to salt intake, to help improve the accuracy of subjects’ salt use recall. 

Similar to table salt, seaweed consumption was also not found to be a positive contributor of iodine status, despite its known influence on status in countries including Korea, Japan and parts of coastal Alaska [50,51]. Our negative correlation between seaweed consumption and iodine status may be due to the low number of participants (*n* = 26) who reported eating seaweed and our use of a single value for seaweed by “serving” (Table 1). As different types of seaweed have different amounts of iodine [33], future versions of the FFQ should consider including better descriptions of the type of seaweed consumed to improve estimated iodine consumption. For example, one analysis of 12 common types of seaweed estimated that nori contained ~16 µg while processed kelp contained over 8000 µg per serving. Additionally, iodine status was not found to differ according to dietary patterns identified on the Dietary Habits Survey (vegetarian status, CSA membership, regular gardener, etc.). This is likely due to a low sample size of participants who identified with these specific dietary habits. Only about half of the subjects identified with at least one dietary preference (*n* = 61). Further studies could target these groups, as well as those adhering to AHA guidelines regarding reduced salt intake, to better analyze the influence of dietary habits on iodine status. 

As both insufficient and excess iodine intake can affect thyroid function, a secondary purpose of this study was to explore the relationship between iodine status and serum TSH concentration as a marker of hypothyroidism and general thyroid function [7]. While we were expecting that participants with 24 h UICs in the WHO severe-to-moderate deficiency category would have high, normal or slightly elevated TSH concentrations, we instead found no relationship between thyroid function and WHO iodine deficiency category. In the typical hormone feedback scenario, serum TSH concentration should increase as iodine status drops [52], although the thyroid is very efficient at compensating for instances of low and excess iodine intake [53] and 24 h UIC may be more reflective of acute intake than long-term status (as discussed below). For example, a healthy adult stores ~15–20 mg of iodine, with 70–80% of these stores in the thyroid. These stores are in excess of the daily recommendation of 150 µg and can help prevent a drop in the synthesis of thyroid hormone (and subsequent increase in TSH) during periods of low iodine intake. The thyroid can also alter the efficiency of iodine organification or the incorporation of iodine into the thyroglobulin in times of excess iodine intake via ‘the Wolff-Chaikoff effect’ [54,55]. The weak but significant association between TSH and 24 h UIC (r = 0.194, *p* = 0.045) could be explained by a few participants that shifted the simple correlation in the positive direction in our sample, which had only a few subjects with TSH concentration falling in the hypo- (TSH >4.68 mIU/L) or hyperthyroid (TSH <0.47 mIU/L) ranges, at 3.7% and 1%, respectively. The data may have also been influenced by the inclusion of participants with autoimmune disorders (as we did not measure thyroid autoantibodies) or the inclusion of four smokers.

The lack of a national database for iodine content in food was a major limitation of this study. The USDA is currently developing an iodine database for commonly consumed food products [56] which should greatly improve the assessment of iodine intake in relation to iodine status. However, the potential variability of the iodine content of foods due to iodine content of the soil in different regions [3] is likely to consistently complicate assessment of iodine intake and status. Further modification of our FFQ may also enhance its future use as a tool to estimate iodine intake and help determine the dietary habits that place an individual at risk of compromised iodine status. These include better clarification of iodized salt and specific fish and seaweed sources. Portion sizes may also be better quantified by allowing participants the opportunity to enter typical portion sizes or by including images and metaphors to visually clarify standard serving sizes (i.e., “3 oz”). A major strength of our study, however, was the creation of our own aggregated dietary iodine database for items in our FFQ. We combined multiple sources of the recent literature investigating iodine content in various foods (Table 1) and converted the listed units to iodine content per serving for better applicability to our FFQ and, ultimately, increased ease of analysis. The significant association between estimated dietary iodine intake from the FFQ and UIC further increased our confidence in our novel database. This database can be built upon for additional research until the USDA completes their official iodine database. 

An additional limitation of our study and many others is the use of UIC as the biomarker of iodine status. Twenty-four hour UIC is a widely used biomarker for the assessment of iodine status, as approximately 90% of dietary iodine is excreted in the urine [57]. However, 24 h UIC is prone to collection and methodological errors and is thought to be more representative of acute (i.e., days) versus chronic iodine status [15], due to variations in 24 h UIC samples [58]. Some of this variability may be due to day-to-day differences in dietary iodine intake, but confounders such as circadian [59] and seasonal differences [60,61] also account for individual variability. Currently, there is no consensus on the biomarker to use for assessment of individual iodine status [22,23]. Additionally, in the current study, differences in the period of collection in our comparison of 24 h UIC and our FFQ, which addressed intake over three months, may have been a further limitation. While other biomarkers for assessment of this status have been suggested, including multiple 24 h UIC samples [22], age-adjusted iodine–creatine ratio [62], and serum thyroglobulin concentration [63], use of these markers are not without error and come with practical and logistics concerns when testing a large number of participants [64]. Thus, as mentioned previously, the results of the current study, particularly those related to UIC, are relevant only to the sample examined and cannot be extrapolated to, and are not necessarily representative of young adults, in our town, the mountain west region or elsewhere in the US. They do, however, support the need for continuous national iodine monitoring with emphasis on subgroups that may be susceptible to iodine deficiency [21] based on dietary choices.

## 5. Conclusions

The current study found an indication that our sample of healthy young individuals living in the mountain west region of the U.S. may be borderline deficient (based on a median 24 h UIC of 98 µg/L), although a high degree of variability in 24 h UIC was observed. These results, however, are indicative of the status of this sample only, and not of other, similar subsamples. Dairy and egg consumption were found to be significant predicators of 24 h UIC, whereas reported intake of iodized salt was not. Iodized salt consumption may either be difficult to assess reliably, or not as big a predictor of iodine status in iodine-replete areas. Development of a national dietary iodine database and better biomarkers for the assessment of iodine status in individuals will greatly improve understanding of the relationships between iodine intake, iodine status and thyroid function. 

## Figures and Tables

**Figure 1 nutrients-12-00121-f001:**
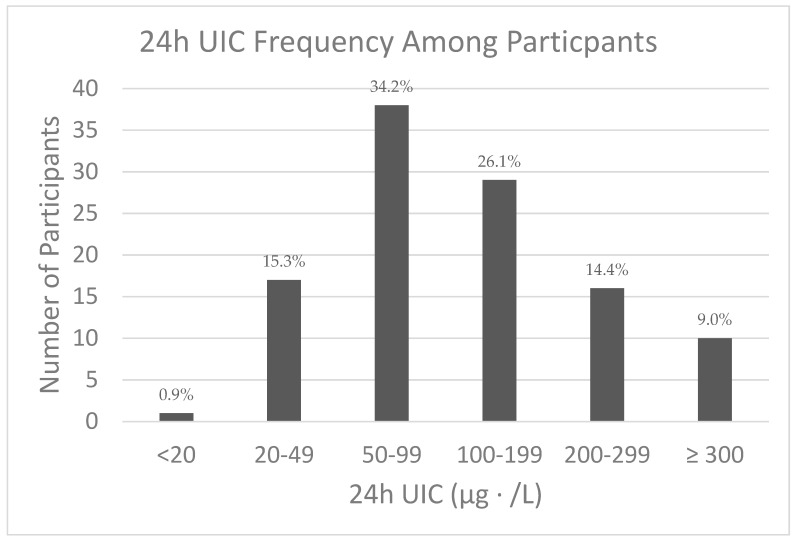
Iodine status based on the World Health Organization (WHO) criteria for urinary iodine concentration (UIC) [15]: <20 µg/L (Severe Deficiency); 20–49 µg/L (Moderate Deficiency); 50–99 µg/L (Mild Deficiency); 100–199 µg/L (Adequate); 200–299 µg/L (More than Adequate); ≥300 µg/L (Excessive).

**Figure 2 nutrients-12-00121-f002:**
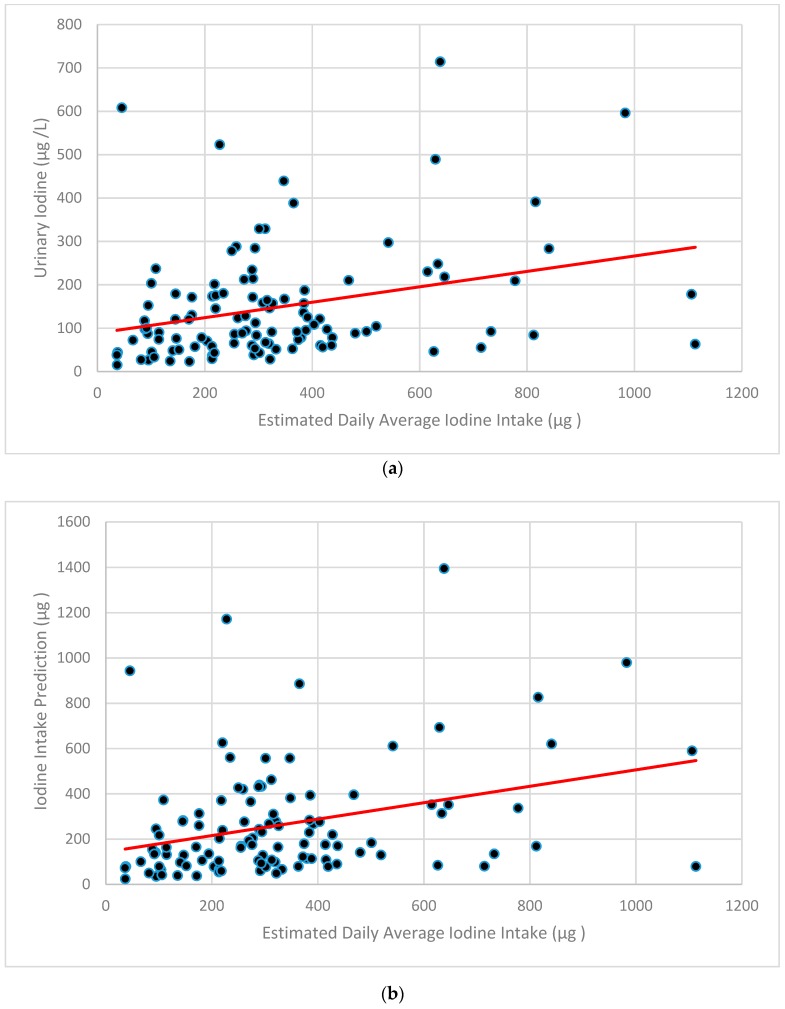
Estimated iodine intake by Food Frequency Questionnaire versus Urinary Iodine Concentration (**a**) and Iodine Intake Prediction (using the equation of Zimmerman) [26,27] where Daily Iodine Intake = Urinary iodine (µg/L) × 0.0235 × body weight (kg) (**b**).

**Table 1 nutrients-12-00121-t001:** Iodine content of foods deduced from current available resources.

Food Item	Estimated Iodine Level (µg per Serving)	Serving Size
Milk (fluid)	90.86 ^a^	1 cup
Soy Milk	2.2 ^b^	1 cup
Soy Protein Bar	20 ^c^	1 bar
Soy Protein Powder	0 ^d^	1 scoop
Soy Sauce	0 ^d^	1 Tbsp
Non-dairy Milk	2.2 ^b^	1 cup
Orange Juice	0 ^a^	1 cup
Cereal	1.62 ^a^	¾ cup
Bread	1.18 ^a^	1 slice (26 g)
Subway Sandwich	4.14 ^a^	6-inch sandwich
Bagel	4.312 ^a^	1 bagel (95 g)
Yogurt	87 ^a^	1 cup
Cheese	13.33 ^a^	2 oz
Egg	21.42 ^a^	1 egg (50 g)
Margarine	0 ^a^	1 tsp
Liver	11 ^a^	100 g (3.5 oz)
Cod	93 ^e^	3.5 oz
Grouper	84 ^f^	3.5 oz
Haddock	224 ^e^	3.5 oz
Halibut	9.9 ^e^	3.5 oz
Herring	84 ^f^	3.5 oz
Mackerel	84 ^f^	3.5 oz
Perch	10.89 ^e^	3.5 oz
Salmon	10.43 ^e^	3.5 oz
Sardines	6.69 ^g^	3.5 oz
Seabass	84 ^f^	3.5 oz
Swordfish	19.8 ^e^	3.5 oz
Tukaoua	84 ^f^	3.5 oz
Tuna Albacore	6.69 ^a^	3.5 oz
Tuna Light	6.69 ^a^	3.5 oz
Walleye	84 ^f^	3.5 oz
Other Fish	22 ^h^	3.5 oz
Clams	74.8 ^e^	4 oz
Crabmeat	42.56 ^e^	4 oz
Lobster	209.67 ^e^	4 oz
Mussels	9.14 ^i^	4 oz
Oysters	135 ^e^	4 oz
Scallops	9.14 ^e^	4 oz
Shimp	8.184 ^a^	4 oz
Seaweed	34.56 ^j^	2.6 g (1 sheet)
Iodized Table Salt	68 ^k^	1.5 g (1/4 tsp)
Other Salt	0 ^l^	1.5 g
Multivitamin-(Iodine-Containing)	150 ^m^	1 tablet
Kelp Supplement	225 ^c^	1 capsule

Sources: ^a^ Total Diet Study 2006–2013 [29], ^b^ Bath et al. [30], ^c^ Popular product nutrient labels, ^d^ American Thyroid Association [31], ^e^ United States Department of Agriculture (USDA) National Food and Nutrient Analysis Program [32], ^f^ Inferred from other fish values from USDA’s National Food and Nutrient Analysis Program, ^g^ Inferred from Canned Tuna TDS 2006–2013, ^h^ Inferred from the average of multiple fish types from USDA’s National Food and Nutrient Analysis Program, ^I^ Inferred from the scallop value from USDA’s National Food and Nutrient Analysis Program, ^j^ Teas et al. 2004 [33], ^k^ Total Diet Study 1982–1991 [34], ^l^ Dasgupta et al. 2008 [35], ^m^ Summation of observation of popular multivitamins available at local grocery stories and pharmacies.

**Table 2 nutrients-12-00121-t002:** Descriptive characteristics of Participants and Frequency of Overweight, Smoking and Physical Activity by Sex.

	Mean ± SEM		*n*
**Age**	31.6 ± 0.8		111
Height (cm)	173.2 ± 1.0		111
Body Mass (kg)	74.9 ± 1.7		111
BMI (kg/M^2^)	24.8 ± 0.4		111
Hematocrit (%)	45.1 ± 0.28		110
TSH (mIU/L)	2.1 ± 0.11		107
	Male (*n*)	Female (*n*)	
Sex	59	52	111
BMI Underweight	1	0	1
BMI Normal	25	36	61
BMI Overweight	26	13	39
BMI Obese	7	3	10
Smokers	4	0	4
IPAQ Low	10	7	17
IPAQ Moderate	21	30	51
IPAQ High	22	14	36

**Table 3 nutrients-12-00121-t003:** Frequency of major sources of reported dietary iodine by serving.

Food Serving Size	Iodine (µg per Serving)	Frequency
Milk (1 cup fluid) (*n* = 110 *)	90.86	0 or <1 times·month^−1^ = 17
		1–3 times·month^−1^ = 12
		1 time·week^−1^ = 10
		2–4 times·week^−1^ = 26
		5–6 times·week^−1^ = 15
		1 time·day^−1^ = 17
		2–3 times·day^−1^ = 11
		4–5 times·day^−1^ = 2
Yogurt (1 cup) (*n* = 111 *)	87	0 or <1 month^−1^ = 26
		1–3 times·month^−1^ = 17
		1 time·week^−1^ = 15
		2–4 times·week^−1^ = 24
		5–6 times·week^−1^ = 18
		1 time·day^−1^ = 9
		2–3 times·day^−1^ = 2
		4–5 times·day^−1^ = 0
Eggs (1 whole) (*n* = 109 *)	21.42	0 or <1 month^−1^ = 16
		1–3 times·month^−1^ = 8
		1 time·week^−1^ = 15
		2–4 times·week^−1^ = 29
		5–6 times·week^−1^ = 18
		1 time·day^−1^ = 6
		2–3 times·day^−1^ = 11
		4–5 times·day^−1^ = 6
Total Seafood (3.75 oz) (*n* = 110 *)	61	0 or <1 month^−1^ = 28
		1–3 times·month^−1^ = 48
		1 time·week^−1^ = 22
		2–4 times·week^−1^ = 9
		5–6 times·week^−1^ = 1
		1 time·day^−1^ = 2
		2–3 times·day^−1^ = 0
		4–5 times·day^−1^ = 0
Iodized Table Salt (1.5 g) (*n* = 108 *)	68	0 or <1 month^−1^ = 28
		1–3 times·month^−1^ = 18
		1 time·week^−1^ = 12
		2–4 times·week^−1^ = 12
		5–6 times·week^−1^ = 12
		1 time·day^−1^ = 14
		2–3 times·day^−1^ = 10
		4–5 times·day^−1^ = 2
Multivitamin (1 tablet) (*n* = 106 *)	150	0 or <1 month^−1^ = 67
		1–3 times·month^−1^ = 8
		1 time·week^−1^ = 3
		2–4 times·week^−1^ = 4
		5–6 times·week^−1^ = 4
		1 time·day^−1^ = 17
		2–3 times·day^−1^ = 2
		4–5 times·day^−1^ = 1

* Number of participants who reported consuming each food source.

**Table 4 nutrients-12-00121-t004:** Backwards Regression Model with Estimated iodine intake from food frequency questionnaire (FFQ) as the dependent variable and estimated iodine intake from various food categories as independent variables.

	R^2^	SEE	Beta	Sig.
Model	0.991	21.81		0.005
Total Dairy	-	-	0.066	>0.001
Total Fish	-	-	0.108	>0.001
Multivitamin	-	-	0.034	>0.001
Iodized Table Salt	-	-	0.033	>0.001
Seaweed	-	-	0.122	>0.001
Egg	-	-	0.032	>0.001

SEE, standard error of the estimate.

**Table 5 nutrients-12-00121-t005:** Backwards regression models with urinary iodine concentration (UIC) as the dependent variable and estimated iodine intake from total dairy, total fish, multivitamin, iodized salt, seaweed and egg consumption entered as independent variables.

	R^2^	SEE	Beta	Sig.
Model 1	0.229	118.1		0.001
Total Dairy	-	-	0.017	0.001
Total Fish	-	-	−0.033	0.249
Multivitamin	-	-	−0.006	0.128
Iodized Table Salt	-	-	0.003	0.704
Seaweed	-	-	0.111	0.455
Egg	-	-	0.037	0.032
Model 2	0.228	117.6		0.001
Total Dairy	-	-	0.017	0.001
Total Fish	-	-	−0.033	0.234
Multivitamin	-	-	−0.006	0.118
Seaweed	-	-	0.105	0.474
Egg	-	-	0.038	0.024
Model 3	0.224	117.3		0.001
Total Dairy	-	-	0.016	0.001
Total Fish	-	-	−0.033	0.234
Multivitamin	-	-	−0.006	0.125
Egg	-	-	0.040	0.016
Model 4	0.213	117.5		0.001
Total Dairy	-	-	0.016	0.001
Multivitamin	-	-	−0.006	0.136
Egg	-	-	0.041	0.013
Model 5	0.197	118.2		0.000
Total Dairy	-	-	0.016	0.000
Egg	-	-	0.039	0.018

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
