# Peer review of "Dietary Relationship with 24 h Urinary Iodine Concentrations of Young Adults in the Mountain West Region of the United States"

_nutrients, 2020, doi:10.3390/nu12010121_

Round 1

Reviewer 1 Report

The article addresses very interesting aspects of the dependence of iodine supply on the diet used. The obtained results may have global significance because it is common, as part of iodine prophylaxis, for salt iodization, which according to the Aurors was not the main source of this element.
I have only minor comments regarding changes in the text:
1 I suggest reducing the number of key words
2 I suggest using volume units compatible with the SI system
3 I suggest replacing the sign / by · in the units (e.g. g / dm3 with the sign g · dm-3)

Author Response

Thanks for the comments of reviewer. We have revised according to your comments.

Reviewer 2 Report

Please see attachment below.

I have not provided separate comments for authors and editors - I am happy for the authors to see all comments made.

Reviewer 3 Report

An interesting report and important for health implications in young people. 

This is a secondary analyses of data, however, I think there should be a sentence or two about ethics/requirement to ensure the reader knows. 

An observation - why is normal weight at 18.5 BMI, usually it is 20BMI, below this is underweight?

Did vegetarians have a higher level of deficiency as those who were not vegetarian? This could skew the results. 

Reviewer 4 Report

Abstract, line 10,

According to the WHO, a median urinary iodine concentration (UIC) of 100-199 μg/L represents the status of adequate iodine nutrition. While median UIC is an indicator of the iodine status of a sampled population, a single UIC value cannot be used to define each individual’s iodine status. The percentage of each UIC category, such as <50 μg/L or 50-100 μg/L etc. describes the distribution within the group. Hence, the statement “50.4% of patients were mildly iodine deficient (UIC<100 μg/L)” is inappropriate. A median UIC should be given.

The same issue should be addressed in other parts of the article. For example, in Figure 1, the categorization should be “UIC <20 μg/L, 20-49 μg/L, 50-99 μg/L….” instead of “severe deficiency, moderate deficiency, mild deficiency…”.

Table 1. Iodine content of milk

Does the estimated iodine level of milk provided here refer to data from liquid milk? Is milk prepared from milk powder included in the current FFQ? The iodine content of milk prepared from milk powder may be different from that of liquid milk.

Table 1. Food item-multivitamin

Does the “multivitamin” here mean “iodine-containing multivitamin”? This should be specified.

Results-3.2. “Mean UIC for participants was 146.9 ± 12.4 mg/L….”

The median UIC and distribution (eg. p25-p75) should be provided.

Results-3.3. “Overall estimated daily iodine intake averaged 327.7 μg/L ± 21.0…”

The median value and distribution (eg. p25-p75) should also be provided.

Results-3.6. Relationship between iodine status and thyroid function

Is there data on thyroid autoantibodies and the smoking status of participants? These factors may interfere with participants’ thyroid function. If these data is lacking, it should be stated as a limitation of current study.

Round 2

Reviewer 2 Report

Please see attached comments
